# Function of Yogurt Fermented with the *Lactococcus lactis* 11/19-B1 Strain in Improving the Lipid Profile and Intestinal Microbiome in Hemodialysis Patients

**DOI:** 10.3390/nu17111931

**Published:** 2025-06-04

**Authors:** Yoshiki Suzuki, Ken Ishioka, Taichi Nakamura, Nozomu Miyazaki, Shigeru Marubashi, Tatsuo Suzutani

**Affiliations:** 1Department of Microbiology, Fukushima Medical University School of Medicine, 1 Hikarigaoka, Fukushima 960-1295, Japan; suzukiyk@fmu.ac.jp (Y.S.); ishiken@fmu.ac.jp (K.I.); m-nozomu@fmu.ac.jp (N.M.); 2Department of Hepato-Biliary-Pancreatic and Transplant Surgery, Fukushima Medical University School of Medicine, 1 Hikarigaoka, Fukushima 960-1295, Japan; s-maru@fmu.ac.jp; 3Department of Nephrology, Mito Red Cross Hospital, 3-12-48 Sannomaru, Mito 310-0011, Ibaragi, Japan; ta-nakamura@mito.jrc.or.jp; 4Vaccine Center, Ohara General Hospital, 6-1 Agemachi, Fukushima 960-8611, Japan

**Keywords:** uremic toxin, *Lactococcus lactis* 11/19-B1, microbiome, hemodialysis, CKD

## Abstract

Background/Objectives: The number of chronic kidney disease (CKD) patients is increasing in Japan, and this population is at high risk of death from cardiovascular and cerebrovascular diseases. Therefore, prevention of arteriosclerosis as a common underlying cause of these diseases is required. In this study, we examined whether 11/19-B1 yogurt, which has been proven to reduce serum low-density lipoprotein (LDL) levels, can decrease the serum levels of indoxylsulfate and trimethylamine-*N*-oxide (TMAO), which are produced by intestinal microbiota and known to cause arteriosclerosis, through improving dysbiosis in hemodialysis patients. Methods: Nineteen dialysis patients consumed 50 g of 11/19-B1 yogurt daily for 8 weeks, and changes in serum lipid profile and uremic toxin levels, intestinal microbiome, as well as the frequency of bowel movement and stool characteristics were observed. Results: The results demonstrated that an intake of yogurt decreased serum LDL 99.3 to 88.5 (*p* = 0.049) and indoxylsulfate in seven of nine subjects with previously high concentrations, and improved stool characteristics as estimated by the Bristle stool score, although decreased HDL and no beneficial effect on serum TMAO was observed. Conclusions: These results may suggest that the ingestion of 11/19-B1 yogurt provides a preventative effect against the progression of atherosclerosis and renal dysfunction.

## 1. Introduction

The increase in chronic kidney disease (CKD) incidence is a major global health problem [1]. CKD can progress to end-stage renal disease (ESRD), resulting in the need for dialysis. Japan is experiencing a declining birthrate and an aging population, with approximately 30% of the population now being classified as elderly. As a result, approximately 1 in 8 adults is estimated to have CKD in Japan, and 340,000 people with ESRD are introduced to dialysis due to a lack of cadaver kidney donations for renal transplantation [2,3]. This situation has resulted in a significant financial burden on the health care system. Therefore, preventing the progression of kidney dysfunction is an urgent issue.

The progression of CKD is associated with arteriosclerosis, which is caused by lifestyle-related diseases such as high cholesterol, high blood sugar, and obesity [1]. Further, uremic toxins are key factors for arteriosclerosis, accelerating the progression of CKD and cardiovascular diseases [4,5]. Although more than 100 substances are classified as uremic toxins, a few uremic toxins, such as indoxyl sulfate, trimethylamine-*N*-oxide (TMAO) and p-cresol, are produced by intestinal microbiota [6,7]. These factors are particularly potent arteriosclerosing agents [8]. Most uremic toxins are excreted into the urine, but their excretion decreases due to renal dysfunction, creating a negative spiral that further accelerates the progression of renal failure [9,10]. Moreover, CKD patients tend to have dysbiosis and suffer from constipation at a relatively high rate [11,12]. These factors indicate that improving dysbiosis to lower uremic toxins is an attractive and important preventive strategy for limiting the progression of CKD [13].

Low-density cholesterol (LDL) is another major risk factor leading to arteriosclerosis [14]. Previously, we reported that the ingestion of 11/19-B1 yogurt containing the *Lactococcus lactis* 11/19-B1 strain (*L. lactis*) and *Bifidobacterium lactis* BB-12 strain (*B. lactis*) for 8 weeks had hypocholesterolemic and immunostimulatory effects in healthy adults [15]. The results suggested that the 11/19-B1 yogurt might have the potential to not only lower LDL but also uremic toxins through the improvement of dysbiosis, resulting in the limitation of CKD progression. Therefore, we examined whether consumption of the yogurt could reduce the serum concentration of uremic toxins, i.e., indoxyl sulfate and trimethylamine-*N*-oxide (TMAO), derived from gut microbiota, in dialysis patients who are unable to excrete uremic toxins. At the same time, we also sought to clarify whether the LDL-lowering and immunostimulating effects could also be detected in dialysis patients.

## 2. Materials and Methods

### 2.1. Subjects and Study Design

This study was approved by the Ethics Committee of Fukushima Medical University (Approval No. REC2022-011) and that of Mito Red Cross Hospital, and was conducted in accordance with the guidelines laid down in the Declaration of Helsinki. Written consent was obtained from all subjects prior to enrollment in this study. This trial was registered as ID number 20230221-182102 in the University Medical Information Network (UMIN; https://www.umin.ac.jp/english, accessed on 23 March 2023).

The dialysis subjects were recruited from the Nephrology Department of Mito Red Cross Hospital. Their characteristics are shown in Table 1. The reason for choosing dialysis patients as subjects in this study is that their serum uremic toxin levels directly reflect the amount produced by gut microbiota. Moreover, as they came to the hospital three times a week for dialysis, we could frequently observe them, and provide immediate treatment if their blood potassium or phosphorus levels rose too high due to the yogurt intake.

After a pre-observation period of 4 weeks, the subjects ingested 50 g of the yogurt per day for the 8-week intervention period, which was followed by an 8-week washout period (Figure 1). Yogurt intake was set at 50 g per day to prevent serum potassium and phosphorus levels becoming excessively high in dialysis patients. Week 16 was the endpoint in this study. Blood tests were performed every 4 weeks, and the serum concentrations of the uremic toxins, indoxyl sulfate and TMAO were evaluated at two time points, week 0 and 8. Stool samples were collected at 3 time points, week 0, 8 and 16. We initially planned to compare week 0 with week 8 (before and after the intervention) and week 8 with week 16 (before and after the wash-out period). One subject was transferred to a different hospital for clinical reasons after week 8, so we compared the data for 19 subjects between week 0 and week 8 and those for 18 subjects between week 8 and week 16. Throughout the study period, the subjects provided a daily record of whether they consumed the yogurt, their bowel movement frequency, stool characteristics, and any subjective symptoms.

### 2.2. Yogurt

The yogurt was fermented with common starters, the *Streptococcus thermophilus* ST-20 strain, *Lactobacillus delbrueckii* subspecies bulgaricus LB-12 strain and *Lactobacillus acidophilus* La-5 strain, as well as the *L. lactis* 11/19-B1 and *B. lactis* BB-12 strains [15,16,17], and was produced once a week by Rakuou Kyodo Milk Industry Co., Ltd. (Fukushima, Japan). The yogurt contained 10^7^ cfu/mL of lactic acid bacteria. It was transported at 4 °C to the Mito Red Cross Hospital, and then distributed to patients at 4 °C for the intervention period.

### 2.3. Blood and Stool Samples

The blood samples were collected in blood collection tubes immediately before the start of dialysis, and serum samples were prepared by centrifugation at 3500 rpm for 7 min. The serum samples were transported at −20 °C to Fukushima Medical University within 48 h, and then stored at −70 °C until analyses.

The stool specimens were collected by the patients using a kit for easy stool collection (FS-0017, TechnoSuruga Lab. Co., Ltd., Shizuoka, Japan) after receiving a lecture on how to properly collect stool samples. The collected stool samples were temporarily stored at Mito Red Cross Hospital at −20 °C within 48 h of collection, and then stored at −70 °C in Fukushima Medical University.

### 2.4. Analyses of Serum Uremic Toxin Concentrations (Indoxyl Sulfate and TMAO)

The concentration of indoxyl sulfate in the serum samples was measured using an indoxyl sulfate assay kit (NIPRO, Tokyo, Japan) according to the instructions provided with the kit. The principle of the kit is that the amount of indoxyl sulfate (μmol/L) in the sample is quantified by (i) converting indoxyl sulfate into indoxyl using sulfatase, and then (ii) measuring the amount of formazan, converted from WST-8 by indoxyl, at an absorbance of 450 nm.

The concentration of trimethylamine-N-oxide (TMAO) was determined by LC-MS at Kyusyu Pro Search LLP (Fukuoka, Japan) as part of commissioned research.

### 2.5. Statistical Analysis

Graphpad prism (version 10.4.1 for Mac, GraphPad Software, San Diego, CA, USA, www.graphpad.com) was used for statistical analysis and to visualize output data. The comparative analysis of blood test biochemistry, uremic toxin and stool characteristics for week 0 and week 8 was undertaken using a paired *t*-test. In the case of gut microbiota, the relative proportion of each taxon outputted by the analysis carried out using Quantitative Insights Into Microbial Ecology2 (QIIME2) version 2024.5 was calculated for each subject [18]. The “Responder (R)” (*n* = 11) was defined as the group in which indoxyl sulfate decreased after the intervention, and the “Non-Responder (NR)” (*n* = 8) was defined as the group in which indoxyl sulfate did not decrease after the intervention. The Wilcoxon signed-rank test was used for comparisons between two corresponding groups. The Mann–Whitney U test was used for comparisons between two groups with no correspondence.

### 2.6. Procedure for 16S rRNA Sequencing

For an analysis of the gut microbiome, bacterial DNA extraction from the stool samples and 16S rRNA gene sequencing were carried out using the previously reported procedure with some modifications [19]. Briefly, stool samples were suspended in 10mM EDTA solution and beaten in the presence of EZ-Beads (AMR Inc., Gifu, Japan) using a MagNA Lyser Instrument (Roche Diagnostics International Ltd., Rotkreuz, Switzerland). The DNA was purified with Phenol and Phenol/Chloroform extraction and stored at −30 °C until use. The V1–V2 regions of the 16S rRNA gene were amplified with PCR using the following primer set: NexXT27F-2, 5′-GTCTCGTGGGCTCGGAGATGTGTATAAGAGACAGagrgtttgatymtggctcag-3′, and NexXT338R-2, 5′-TCGTCGGCAGCGTCAGATGTGTATAAGAGACAGtgctgcctcccgtaggagt-3′.The PCR was carried out as follows: initial denaturation at 95 °C for 3 min was followed by 20 cycles of denaturation at 95 °C for 30 s, annealing at 55 °C for 30 s, extension at 72 °C for 30 s, and a final extension at 72 °C for 5 min. The PCR products of about 400 bp were purified with Agencourt AMPure XP magnetic beads (Beckman Coulter, Brea, CA, USA) on FastGene MagnaStand 96 (Nippon Genetics, Tokyo, Japan). The PCR products were attached with the dual index and sequencing adaptors by via second PCR using the Nextera XT Index Kit (Illumina, San Diego, CA, USA). The second PCR program consisted of the following steps: initial denaturation at 95 °C for 3 min, 8 cycles at 95 °C for 30 s, 55 °C for 30 s, and 72 °C for 30 s, and a final extension at 72 °C for 1 min. After purification, the concentration of the PCR products was determined with a Quantus Fluorometer (Promega, Madison, WI, USA) and equally mixed in molar concentrations to generate a 4 nM library pool. The 16S rRNA gene libraries were sequenced with 2 × 250 bp paired-end reads on the MiSeq system (Illumina, San Diego, CA, USA) for next-generation sequencing using a MiSeq v2 reagent kit (Illumina, San Diego, CA, USA).

### 2.7. Gut Microbiome Statistical Analysis

The raw sequence data outputted by the next-generation sequencer were imported into QIIME2 version 2024.5 [18]. The primer sequence was removed using the Cutadapt plugin version 4.9 in QIIME2 [20]. The trimmed data were denoised to perform sequence quality control with the DADA2 plugin [21] and a feature table with amplicon sequence variants (ASVs) was generated. The quality scores were selected for portions with a median value of 30 or higher. The phylogenetic tree was generated using the align-to-tree-mafft-fasttree plugin included in QIIME2. The sampling depth parameter was set at the minimum number of reads in the samples to maximize the number of samples. Thereafter, alpha diversity (Shannon’s diversity) and beta diversity (Weighted UniFrac distance [22]) were analyzed using the q2-diversity plugin. For the analysis of alpha diversity, numerical data were exported and reanalyzed by Graphpad prism version 10.4.1. The taxonomic classification was performed using the q2-feature-classifier’s classify-sklearn naïve Bayes taxonomy classifier [23] trained on the Silva version 138.1 database processed by the RESCRIPt (Reference Sequence annotation and CuRation Pipeline) plugin [24,25,26].

## 3. Results

### 3.1. Effect of the 11/19-B1 Yogurt on Serum Findings

The characteristics of subjects are shown in Table 1. The distribution of causes of kidney failure are shown in Figure 2.

Significant mean reductions were observed in LDL (*p* = 0.049), high-density lipoprotein (HDL; *p* = 0.035) and total cholesterol (TC; *p* = 0.0088). However, there was no significant change observed in the ratio of LDL to HDL (Figure 3A–D).

The mean value of indoxyl sulfate tended to decrease with the 11/19-B1 yogurt intervention, although there were no significant differences. In seven of the nine subjects with a high concentration of indoxyl sulfate of more than 4.0 μmol/L before yogurt intake, a decrease in indoxyl sulfate was observed (Figure 3E). However, there was no decrease in TMAO values after yogurt consumption (Figure 3F).

In this study, the amount of yogurt consumed was limited to 50 g due to concerns about increases in serum potassium and phosphorus levels in patients. However, no such increases were observed (Figure A2A,B), with only 1 of the 19 patients requiring an increase in the dose of medication to lower blood phosphorus. Further, no serious adverse events were observed.

### 3.2. Effect of 11/19-B1 Yogurt Intake on Stool Characteristics

No significant differences were found in the paired *t*-test for the average number of bowel movements per day (Figure 4A). However, there was an increase in the percentage of patients with stools classified as normal, with scores between 3 and 5 on the Bristle Stool Scale (BSS) (Figure 4B). The percentage of subjects with relatively soft stools (score of 5 or higher) decreased from 15.8% to 0%, while those with hard stools (score of 3 or lower) decreased from 21% to 5.3%, resulting in an increase from 63.2 to 94.7% in patients with stools classified as normal (score ranging between 3 and 5). The chi-square test showed a significant relationship between time and mean BSS (X^2^ = 6.2, *df* = 2, *p* = 0.045), suggesting that the intervention influences the distribution of mean BSS.

### 3.3. Gut Microbiome

We divided the patients into two groups and analyzed the gut microbiota; the group in which indoxyl sulfate was decreased was named the Responder (R) group, and the other group in which it was not decreased was named the Non-Responder (NR) group.

In terms of alpha diversity, although there was no significant difference between week 0 (W0) and week 8 (W8), the comparison between R-W0, NR-W0, R-W8, and NR-W8 showed a significant difference for the Shannon index (Figure 5A). The Responder (R-W0, R-W8) group showed a decrease in the median value after intervention (*p* = 0.032), while the Non-Responder group showed an increase in the median value (*p* = 0.016) (Figure 5B). We also performed a comparison between each group for observed features and Faith’s PD analysis, but no significant differences were observed.

With regard to beta diversity (Weighted UniFrac distance), principal component analysis (PCA) showed that the groups tended to cluster together (Figure 5C,D), but the PERMANOVA test showed no significant differences between groups.

The relative abundance of taxa is shown on a bar graph, together with the results of the statistical tests. Significant differences were observed in five species and three genera in the comparison among the four groups (R-W0, NR-W0, R-W8, and NR-W8) (Figure 5E–L).

## 4. Discussion

The final goal of this study is to help develop functional foods that can prevent chronic kidney, cardiovascular and cerebrovascular diseases, as these diseases account for 25% of deaths in Japan [27]. We have already demonstrated that consuming 11/19-B1 yogurt for 8 weeks can reduce slightly elevated LDL levels and enhance cellular immunity in healthy adults [15]. Therefore, in this study, we investigated whether this yogurt can also reduce uremic toxins derived from gut bacteria, i.e., indoxyl sulfate and TMAO, which are known to be strong causative agents in arteriosclerosis [4,6,28,29]. As a result, the consumption of the yogurt was observed to improve bowel movements and stool consistency, and led to a reduction in LDL and indoxyl sulfate, but not TMAO, indicating the possibility that the yogurt can prevent the progression of arteriosclerosis. However, it remains unclear which specific bacterial strains were responsible for the observed effects, due to the limited number of subjects and the absence of a control group using yogurt.

Uremic toxins are a group of over 100 toxic substances that accumulate in the blood due to impaired kidney function. Only a few uremic toxins are produced from gut bacteria; therefore, the concentrations of three toxins in the serum were determined from their production in the intestine and their excretion from the kidneys. In this study, dialysis patients were chosen as subjects based on the hypothesis that the concentration of these toxins in the serum of dialysis patients, who lack the ability to excrete uremic toxins, reflects their production in the intestines alone. The results showed that 11/19-B1 yogurt has the ability to reduce serum indoxyl sulfate levels, particularly in subjects in whom the level was already relatively high prior to yogurt intake. Moreover, the yogurt improved bowel movements and the consistency of stools to a favorable state in dialysis patients who tended to suffer from constipation [11]. These results suggest that the bacteria included in the yogurt improved dysbiosis and reduced the production of indole in the intestines. However, there were no clear changes in intestinal flora between before and after the yogurt intake even in the Responder group; therefore, we could not identify the bacterial species associated with the reduction in indole synthesis.

Further, we could not identify the bacterial species or the change in intestinal flora related to the improvements in bowel movements and the reduction in uremic toxins as mentioned above, but did observe beneficial effects regarding the gut microbiome when the yogurt was consumed. The median relative abundances of *B. flagilis*, *R. gnavus*, *B. stercoris*, *B. ovatus*, *E. ramosum* and genus *Erysipelatoclostridium* were significantly higher in the Non-Responder group, while those of genus *Agathobactor* and *Faecalibacterium* were higher in the Responder group. *B. flagilis* is associated with the production of uremic toxins such as indoxyl sulfate and p-cresol [30], and *B. ovatus* is also associated with the production of indole [31,32]. A previous report noted that these bacterial species were more common in hemodialysis patients than in healthy individuals [33], although there are no reports linking them to CKD or the production of uremic toxins with regards to the other three species. On the other hand, genus *Agathobacter*, found to be a highly abundant bacterial species in the flora in the Responder group, is related to the production of short chain fatty acids (SCFAs) [34]. SCFAs have an important role as a protective factor for kidney health [35]. Patients with impaired kidney function are advised to limit their intake of vegetables and fruits to prevent elevated serum potassium levels because abnormal serum potassium levels are associated with their mortality [36,37,38]. Under these conditions, it is necessary to devise ways to use prebiotics such as dietary fiber and oligosaccharides to ensure the sufficient growth of lactic acid bacteria. In summary, even if no changes are observed in the microbiota analysis, it is possible to alter the indole production capacity of the intestinal environment through the use of probiotics.

We reported that the consumption of 11/19-B1 yogurt reduced slightly higher serum LDL levels and enhanced cellular immunity, with the former function induced by *L. lactis* 11/19-B1 and/or *B. lactis* BB-12 strains [15]. The mechanism by which these strains lower serum LDL levels remains unknown, but this effect has also been reported for a limited number of strains, i.e., *Lactobacillus acidophilus ATCC 4356* strains [39], the *Lactobacillus plantarm* strain [40], the *Lactobacillus reuteri* strain [40], the *Lactobacillus plantarum S9* strain [41], the *Lactobacillus plantarum 9-41-A* strain and the *Lactobacillus fermentum M1-16* strain [42]. On the other hand, the immunostimulating effects of lactic acid bacteria were observed in many strains, as the same effect was recognized in the control yogurt fermented with *S. thermophilus*, *L. bulgaricus* and *L. acidophilus* in our previous study [15]. However, we expected the *L. lactis* 11/19-B1 strain to have a special function, as the JCM5805 strain, which is of the same bacterial species, has been shown to have the unique ability to secrete interferon from dendritic cells [43,44]. In fact, we have demonstrated that the intake of the live 11/19-B1 strain activates human immune function and improves atopic dermatitis, and that these effects can also be achieved with the intake of heat-killed bacteria in mouse experiments [15,16]. However, no immuno-activating effect was observed in the dialysis patients. We hypothesized the following reasons for this. First, the immune function of dialysis patients may be too low to be activated by yogurt consumption, as can be seen from the fact that infections are the second leading cause of death in dialysis patients. A second possibility is that in previous studies with healthy individuals, 80 g of the yogurt was consumed, but in this study, the intake was limited to 50 g due to concerns about an increase in serum potassium and phosphate levels. However, none of patients developed hyperkalemia and/or hyperphosphatemia; therefore, this yogurt can be safely consumed by dialysis patients, and we intend to conduct a future study with an increase in the amount of yogurt consumption.

It has been revealed that the gut microbiota changes as renal function deteriorates [45]. The purpose of this study was to develop functional foods to prevent the progression of CKD. However, the results showed that the functionality of the yogurt varies with the existing microbiota prior to yogurt consumption, indicating the results described in this paper cannot be simply applied to individuals whose kidney function is preserved. Furthermore, the mechanisms underlying the observed phenomena have not yet been elucidated. These remain a limitation of this study, and further clinical research involving more volunteers with mildly impaired kidney function is necessary.

## 5. Conclusions

Uremic toxins are produced by intestinal bacteria and are responsible for progressive renal dysfunction. Intake of the yogurt fermented with *L. lactis* 11/19-B1 and *B. lactis* BB-12 strains decreased serum indoxyl sulfate and LDL, but also reduced HDL levels. *B. flagilis*, *R. gnavus*, *B. stercoris*, *B. ovatus*, *E. ramosum* and genus *Erysipelatoclostridium* were suggested to be resistance factors for this intervention, while genus *Agathobactor* and genus *Faecalivacterium* were suggested to be favorable factors. The 11/19-B1 yogurt appears to have potential as a prebiotic to suppress the production of uremic toxins, although further investigation is needed to determine whether the same effects can be observed in non-dialysis individuals.

## Figures and Tables

**Figure 1 nutrients-17-01931-f001:**
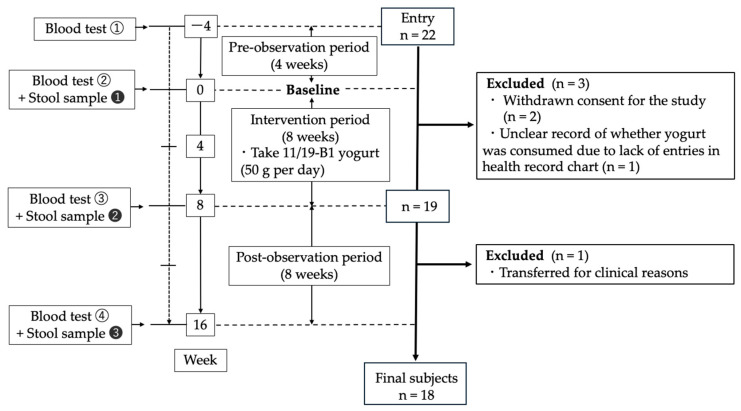
A flowchart of the study design. The white circled numbers represent the blood test number, and the black circled numbers represent the numbers of stool sample collected.

**Figure 2 nutrients-17-01931-f002:**
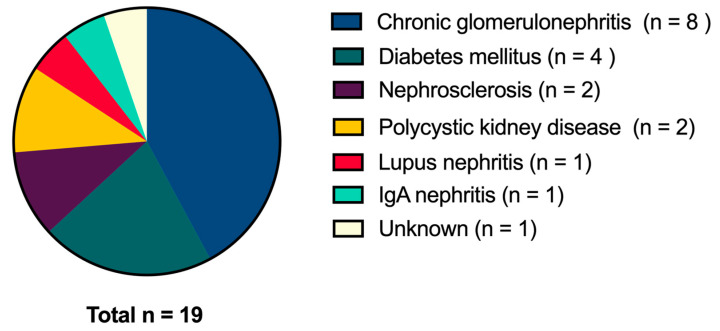
Distribution of causes of kidney failure.

**Figure 3 nutrients-17-01931-f003:**
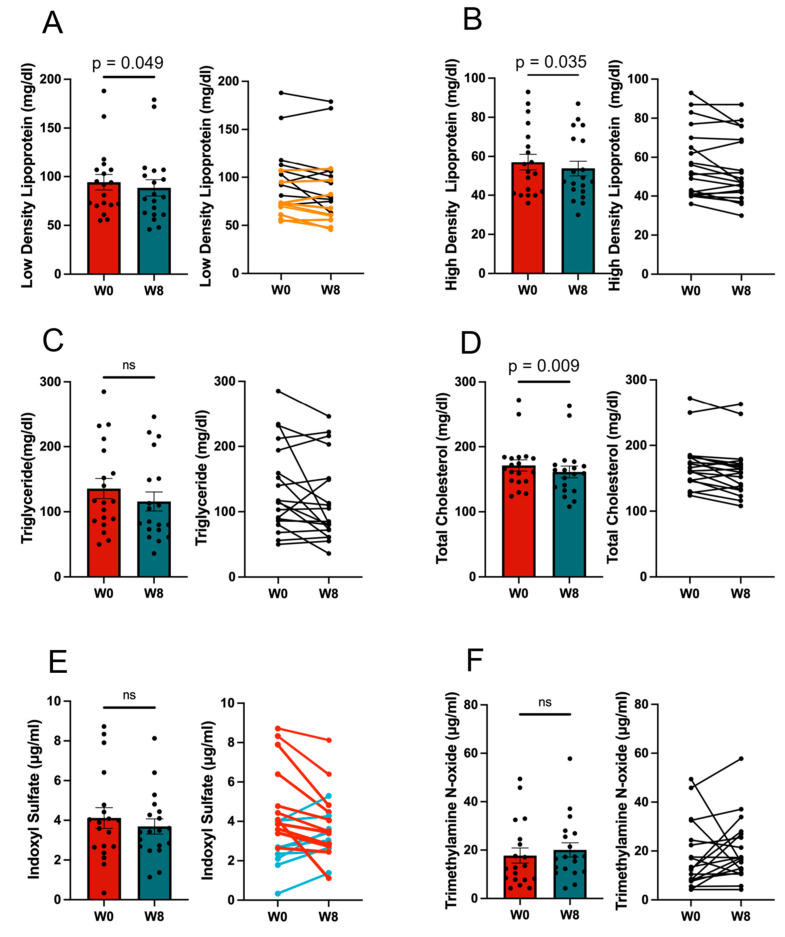
Blood test results for lipid profile and serum uremic toxins before and after 11/19-B1 yogurt intake. (**A**) Low-density lipoprotein (LDL), (**B**) high-density lipoprotein (HDL), (**C**) triglyceride, (**D**) total cholesterol, (**E**) serum indoxyl sulfate and (**F**) serum trimethylamine *N*-oxide (TMAO). The values for each sample were compared using a corresponding *t*-test between week 0 (W0) and week 8 (W8). The bar graphs on the left of each figure represent with mean ± SEM and the graphs on the right show the changes in values for each individual participant. The orange lines in the graph on the right of subfigure (**A**) show the subjects taking statins. The red and light blue lines in the graph on the right of (**E**) represent the subjects in whom serum indoxyl sulfate was decreased (Responder; R) or increased (Non-Responder; NR) after the intervention, respectively. ns—not significant.

**Figure 4 nutrients-17-01931-f004:**
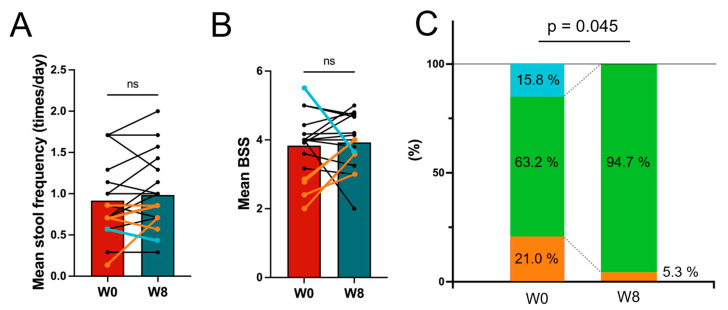
Analysis of stool frequency and stool characteristics. (**A**) The average stool frequency per day. The light blue line indicates the trend for subjects with an average Bristle Stool Scale (BSS) score of less than 3 at W0, and the orange lines for those with an BSS score greater than 5. (**B**) The average BSS score. The light blue line indicates the trend for subjects with an average BSS score of less than 3 at W0, and the orange lines for those with an BSS score greater than 5. (**C**) the average BSS percentage in each week. In each figure, the number for W0 and W8 was calculated from the data from W−1 to W0, and from W7 to W8, respectively. The orange indicates the subject with a BSS score of less than 3 at W0, green indicates with a score between 3 and 5, and light blue indicates a score greater than 5. The chi-square test showed a significant relationship between time and mean BSS. ns—not significant.

**Figure 5 nutrients-17-01931-f005:**
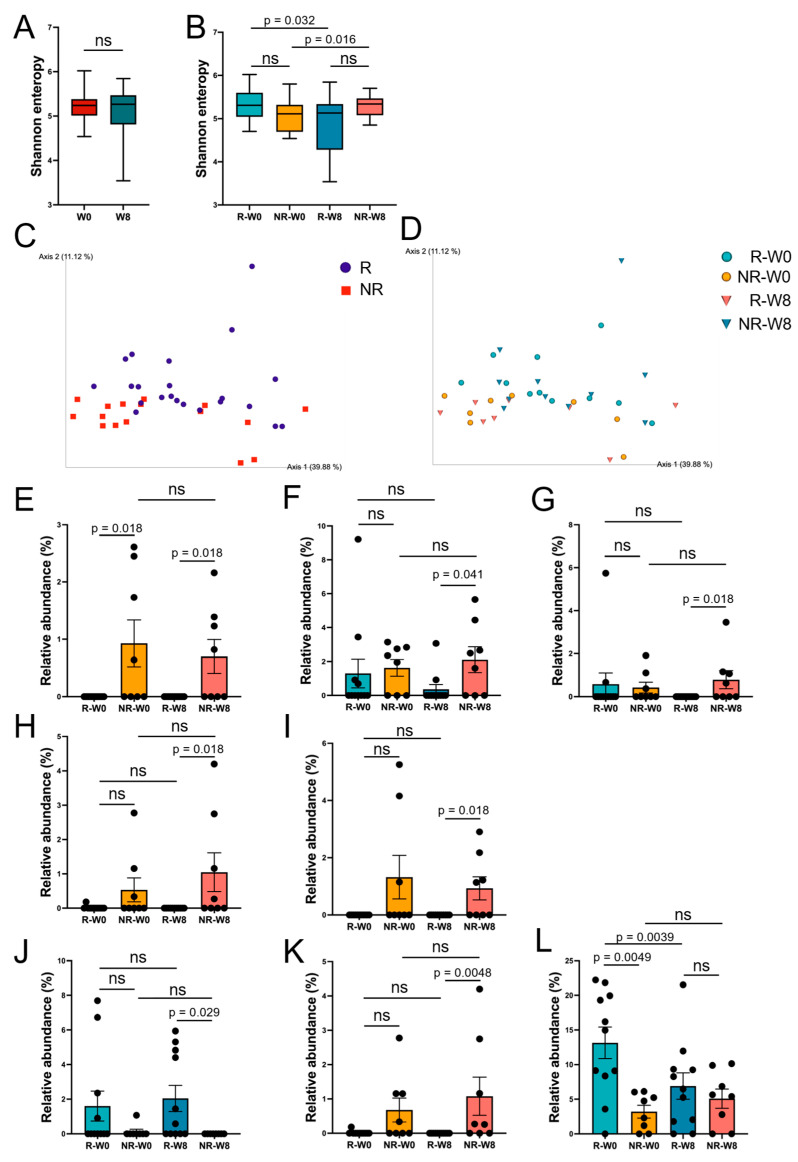
Analysis of gut microbiota and relative abundance of selected taxa. (**A**) Shannon index, W0 vs. W8; (**B**) Shannon index, R-W0 vs. R-W8 vs. NR-W0 vs. NR-W8; (**C**) Weighted UniFrac distance. Principal component analysis (Display R and NR) and (**D**) Weighted UniFrac distance. Principal component analysis (Display R-W0, NR-W0, R-W8 and NR-W8) of the relative abundance of (**E**) Bacteroides flagilis, (**F**) Bacteroides stercoris, (**G**) Bacteroides ovatus, (**H**) Erysipelactoclostridium ramosum, (**I**) Ruminococcus gnavus, (**J**) genus Agathobacter, (**K**) genus Erysipelatoclostridium, and (**L**) genus Faecalibacterium. The bar plots represent mean ± SEM (standard error). The Mann–Whitney U test, which is a nonparametric test without correspondence, was used between R-W0 and between NR-W0 and NR-W0 and NR-W8. Otherwise, the Wilcoxon matched-pairs signed-rank test was used between R-W0 and R-W8 and between NR-W0 and NR-W8, as these were comparisons between the same samples. ns—not significant.

**Table 1 nutrients-17-01931-t001:** Summary of dialysis patients.

	Dialysis Patients Group (N = 19)	
Age	year	65.2 ± 17.6
BMI	kg/m^2^	22.1 ± 4.3
Sex	% of male	58.9
Dialysis period	year	7.5 ± 7.4
Height	cm	162.6 ± 9.1
Body weight	kg	58.4 ± 11.2

Data are presented as mean ± standard deviation.

## Data Availability

The data presented in this study are openly available in Fukushima Medical University at https://www.fmu.ac.jp.

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
