# Peer review of "Function of Yogurt Fermented with the Lactococcus lactis 11/19-B1 Strain in Improving the Lipid Profile and Intestinal Microbiome in Hemodialysis Patients"

_nutrients, 2025, doi:10.3390/nu17111931_

Round 1
Reviewer 1 Report
Comments and Suggestions for Authors
The study of Suzuki et al. investigated whether the ingestion of Lactococcus 11/19-B1 yogurt could provide a preventative effect against the progression of atherosclerosis and renal dysfunction. Overall, although the study was simple, but the findings were interesting and provided useful information in this field. The study was well-designed and the results were clear and could support the conclusion. The current form of manuscript could be considered for publication after only minor revisions.
- In the abstract, citation(Nishiyama et al. 2018) should be deteled.
- For the yogurt prepared in this study, the content of L. lactis 11/19-B1 should be determined.
- How could the authors exclude the effect of Streptococcus thermophilus ST-20 on the results of this study.
- In the conclusion, the limit of the lack of mechanism investigation of this study should be mentioned.
Author Response
Comments 1: In the abstract, citation(Nishiyama et al. 2018) should be deteled.
Response 1: We have deleted the citation from the abstract.
Comments 2: For the yogurt prepared in this study, the content of L. lactis 11/19-B1 should be determined.
Response 2: The total number of lactic acid bacteria has been recorded in the text (L. 101) Unfortunately, the bacterial count by species and the potassium and phosphorus contents were not measured.
Comments 3: How could the authors exclude the effect of Streptococcus thermophilus ST-20 on the results of this study.
Response 3: In our previously reported study, yogurt made with three bacterial strains did not exhibit any LDL-lowering effect. However, when Bifidobacterium lactis and Lactococcus lactis were added to create a yogurt with five strains, an LDL-lowering effect was observed. Based on this result, we reported that one of the two added strains might be responsible for the LDL-lowering effect. However, in this study, the number of subjects was small and no control group was established; all participants consumed yogurt containing the five strains. Therefore, it remains unclear which strain is actually responsible for the observed effect. This limitation has been clearly stated in the main text (L. 262-264).
Comments 4: In the conclusion, the limit of the lack of mechanism investigation of this study should be mentioned.
Response 4: We have added a sentence (L.332-334).
Reviewer 2 Report
Comments and Suggestions for Authors
Abstract - sentence in lane 24 needs rephrasing;
conclusion may be overly ambitious, it seems you have only confirmed previously established knowledge on the effects on LDL, I would rephrase it may provide a preventative effect as atherosclerosis was not directly evaluated
line 47 and 136 246 281 need technical editing
I would put Table 1 in results section, it also needs clarification on data presentation in foot notes i.e. data is presented as mean+/- standard deviation; height is given in cm, not m; in Kg, k should be a lower case letter
Something seems off with indents in this manuscript, please check. On page 3 you need to justify your text - technical editing
As you state in the Manuscript In this study, the amount of yogurt consumed was limited to 50 g due to concerns about increases in serum potassium and phosphorus levels in patients. - so I would like you to give exact recommendations for clinicians and patients based on the results of your study as sole consumption of this product seems risky for the patients and should not be encouraged or?
Figure 2 needs data presentation explanation; figure 2D round p value to 3 decimal spaces
Lowering of HDL should also be taken into account when commenting on the beneficial results of LDL lowering, I am not sure if your interpretation is correct
Figure 4 can be published as supplementary data for better resolution
Conclusion should reflect the results of HDL levels as well for correct interpretation by readers as well the potassium and phosphorous risks for these patients as you can make claims only for the study population and assume it may but may not have effect on different populations
Author Response
Comments 1: conclusion may be overly ambitious, it seems you have only confirmed previously established knowledge on the effects on LDL, I would rephrase it may provide a preventative effect as atherosclerosis was not directly evaluated
Response 1: Thank you very much for your valuable comment. As you rightly pointed out, the LDL-lowering effect is a replication study, and we did not directly observe the prevention of atherosclerosis. However, regarding the LDL-lowering effect, we would like to include the result in abstract considering that the yogurt intake in this study was lower (50g) than that in our previous study (80g). As for the prevention of atherosclerosis, as it is merely a potential effect that we are hoping for, we have intentionally weakened the description (Line 25).
Comments 2: line 47 and 136 246 281 need technical editing
Response 2: Thank you very much for your suggestion. I have made corrections to the inaccurate information (L. 134, 253-255, 290).
Comments 3: I would put Table 1 in results section, it also needs clarification on data presentation in foot notes i.e. data is presented as mean+/- standard deviation; height is given in cm, not m; in Kg, k should be a lower case letter
Response 3: Table 1 has been relocated to the Results section. Additionally, the careless errors have been corrected. Thank you for your valuable suggestions.
Comments 4: Something seems off with indents in this manuscript, please check. On page 3 you need to justify your text - technical editing
Response 4: We have justified the indents as requested.
Comments 5: As you state in the Manuscript In this study, the amount of yogurt consumed was limited to 50 g due to concerns about increases in serum potassium and phosphorus levels in patients. - so I would like you to give exact recommendations for clinicians and patients based on the results of your study as sole consumption of this product seems risky for the patients and should not be encouraged or?
Response 5: No restrictions have been placed on milk consumption in dialysis patients in our hospital, and the intake of yogurt has not resulted in any increase in serum potassium or phosphorus levels, which had been a concern. Therefore, we believe that the consumption of yogurt can be recommended for these patients. However, in order to determine the appropriate amount of yogurt that can be safely consumed, further investigation with increased intake is warranted. To address this point, we have added a sentence to the Discussion section (L. 324-326).
Comments 6: Figure 2 needs data presentation explanation; figure 2D round p value to 3 decimal spaces Lowering of HDL should also be taken into account when commenting on the beneficial results of LDL lowering, I am not sure if your interpretation is correct
Response 6: We have added explanations in Fig. 2. Also the P value in Fig. 2D was rounded to 3 decimal spaces. As you pointed out, the decrease in HDL also needs to be taken into consideration. We have mentioned this point in the main text (L. 24, 340). Thank you for your valuable comment.
Comments 7: Figure 4 can be published as supplementary data for better resolution
Response 7: I agree that the data in Fig. 4 do not directly explain the reduction in serum indoxyl sulfate level, as you pointed out. However, as there are results worth discussing, such as those related to Bacteroides and short-chain fatty acid-producing bacteria, I would like to keep this figure in the manuscript as it is.
Comments 8: Conclusion should reflect the results of HDL levels as well for correct interpretation by readers as well the potassium and phosphorous risks for these patients as you can make claims only for the study population and assume it may but may not have effect on different populations
Response 8: We have added a description regarding the points related to HDL and whether the results of this study can be extended to non-dyalysis individuals (L. 340, 345). Thank you for your feedback."
Reviewer 3 Report
Comments and Suggestions for Authors
Dear Authors, thank you for Your work. The topic is interesting and I read it with attention. Lines 33 and 37 need to be completed with all the mechanisms.
The sample size is really poor so that all the results are not fully reliable. Please enrich the sample size, by including CKD patients deriving equally, if it is possible, from all the causes of CKD thus equalizing the distribution.
Discussion need to be empowered with the mechanisms by wich such a supplementation could help these patients.
Author Response
Comments 1: Lines 33 and 37 need to be completed with all the mechanisms.
Response1: We added one sentence “Japan is experiencing a declining birthrate and an aging population, with approximately 30% of the population now classified as elderly. (L. 32-33)“ and completed the lines highlighted with all the mechanisms related to the financial burden.
Comments 2: The sample size is really poor so that all the results are not fully reliable. Please enrich the sample size, by including CKD patients deriving equally, if it is possible, from all the causes of CKD thus equalizing the distribution.
Response 2: Your point is absolutely valid. As this study was conducted at a single facility, the number of subjects was limited. Taking your comments into account, we would like to conduct a more detailed study with a larger number of participants in our next project (L. 332-335). Thank you very much for your valuable feedback.
Comments 3: Discussion need to be empowered with the mechanisms by which such a supplementation could help these patients.
Response 3: In this study, dialysis patients were selected as subjects to investigate the effect of the yogurt on reducing indoxyl sulfate levels. However, as impaired kidney function cannot be fully restored, the ultimate goal of this study was to prevent the progression of arteriosclerosis in patients with chronic kidney disease whose renal function has not yet significantly deteriorated. Therefore, we added a statement in the conclusion emphasizing the need for research to clarify whether the same effects can be observed in non-dialysis individuals (L. 332-334, 345).